# Dynamics and Predictors of Mortality Due to Candidemia Caused by Different *Candida* Species: Comparison of Intensive Care Unit-Associated Candidemia (ICUAC) and Non-ICUAC

**DOI:** 10.3390/jof7080597

**Published:** 2021-07-24

**Authors:** Yong Jun Kwon, Eun Jeong Won, Seok Hoon Jeong, Kyeong Seob Shin, Jeong Hwan Shin, Young Ree Kim, Hyun Soo Kim, Young Ah Kim, Young Uh, Taek Soo Kim, Jae Hyeon Park, Jaehyeon Lee, Min Ji Choi, Seung A. Byun, Soo Hyun Kim, Jong Hee Shin

**Affiliations:** 1Department of Laboratory Medicine, Chonnam National University Medical School, Gwangju 61469, Korea; garei09@gmail.com (Y.J.K.); Parasite.woni@jnu.ac.kr (E.J.W.); minji1246@naver.com (M.J.C.); tmddk777@naver.com (S.A.B.); alpinboy@chonnam.ac.kr (S.H.K.); 2Department of Laboratory Medicine and Research Institute of Bacterial Resistance, Yonsei University College of Medicine, Seoul 03722, Korea; kscpjsh@yuhs.ac; 3Department of Laboratory Medicine, Chungbuk National University College of Medicine, Cheongju 28644, Korea; ksshin@chungbuk.ac.kr; 4Department of Laboratory Medicine and Paik Institute for Clinical Research, Inje University College of Medicine, Busan 47392, Korea; jhsmile@paik.ac.kr; 5Department of Laboratory Medicine, Jeju National University Medical School, Jeju 63243, Korea; namu8790@jejunu.ac.kr; 6Department of Laboratory Medicine, Hallym University Dongtan Sacred Heart Hospital, Hallym University College of Medicine, Hwaseong 18450, Korea; hskim0901@empas.com; 7Department of Laboratory Medicine, National Health Insurance Service Ilsan Hospital, Goyang 10444, Korea; yakim03@gmail.com; 8Department of Laboratory Medicine, Yonsei University Wonju College of Medicine, Wonju 26426, Korea; u931018@yonsei.ac.kr; 9Department of Laboratory Medicine, Seoul National University College of Medicine, Seoul 03080, Korea; md.kim.taeksoo@gmail.com (T.S.K.); bjack9@gmail.com (J.H.P.); 10Department of Laboratory Medicine, Jeonbuk National University Medical School, Jeonju 54907, Korea; Laboratorymedicine@gmail.com

**Keywords:** ICU, candidemia, mortality, *Candida* species, lack of antifungal therapy

## Abstract

We investigated mortality and predictors of mortality due to intensive care unit-associated candidemia (ICUAC) versus non-ICUAC by *Candida* species. This study included all candidemia cases in 11 hospitals from 2017 to 2018 in South Korea. The all-cause mortality rates in all 370 patients with ICUAC were approximately twofold higher than those in all 437 patients with non-ICUAC at 7 days (2.3-fold, 31.1%/13.3%), 30 days (1.9-fold, 49.5%/25.4%), and 90 days (1.9-fold, 57.8%/30.9%). Significant species-specific associations with 7- and 30-day ICUAC-associated mortality were not observed. Multivariate analysis revealed that ICU admission was an independent predictor of *Candida glabrata* (OR, 2.07–2.48) and *Candida parapsilosis*-associated mortality (OR, 6.06–11.54). Fluconazole resistance was a predictor of *C. glabrata*-associated mortality (OR, 2.80–5.14). Lack (less than 3 days) of antifungal therapy was the strongest predictor of 7-day mortality due to ICUAC caused by *Candida albicans* (OR, 18.33), *Candida tropicalis* (OR, 10.52), and *C. glabrata* (OR, 21.30) compared with 30- and 90-day mortality (OR, 2.72–6.90). *C. glabrata* ICUAC had a stronger association with lack of antifungal therapy (55.2%) than ICUAC caused by other species (30.6–36.7%, all *p* < 0.05). Most predictors of mortality associated with ICUAC were distinct from those associated with non-ICUAC and were mediated by *Candida* species.

## 1. Introduction

Candidemia is an opportunistic infection associated with high morbidity and mortality rates [1,2,3]. In recent decades, the epidemiology of candidemia has greatly changed globally [4]. Although *Candida albicans* is the most common species isolated, the proportions of candidemia due to non-*albicans Candida* species (NAC), such as *Candida glabrata*, *Candida tropicalis*, and *Candida parapsilosis*, continue to increase [5], and the emergence of antifungal resistance in NAC bloodstream isolates has become a matter of concern [6]. Candidemia in intensive care unit (ICU) patients (ICU-associated candidemia, hereafter ICUAC) is an important issue due to its increasing incidence and higher mortality rate than non-ICUAC [7,8,9,10]. Given that each *Candida* species has a unique virulence potential, antifungal susceptibility pattern, and clinical characteristics [5], the changing epidemiology of candidemia may have different implications for the management of—and mortality due to—ICUAC and non-ICUAC according to *Candida* species.

The data on 7, 30, and 90-day mortality rates and predictive factors of ICUAC-associated mortality due to different *Candida* species compared with those of non-ICUAC have not yet been elucidated in detail. In addition, despite recent advances in the antifungal management of ICU patients at high risk of invasive candidiasis [11,12], questions on the impact of lack of antifungal therapy on the *Candida* species-related mortality of ICUAC remain. Therefore, we performed a multicenter study of the epidemiology of candidemia and the antifungal resistance profiles of *Candida* bloodstream isolates at 11 hospitals in South Korea over a 2-year period. In this large population of patients with candidemia, we investigated the clinical features, therapeutic practices including lack of antifungal therapy, and all-cause mortality rates according to *Candida* species in ICUAC vs. non-ICUAC patients. We also assessed the clinical and microbiological variables associated with mortality due to ICUAC vs. non-ICUAC caused by four common *Candida* species. This is the first contemporary study to show significant differences in 7-, 30- and 90-day mortality rates and predictors of mortality between patients with ICUAC caused by four common *Candida* species and those with non-ICUAC in South Korea.

## 2. Materials and Methods

### 2.1. Candida Isolates and Clinical Data Collection

This multicentre study was designed to investigate all cases of *Candida* bloodstream infections (BSIs) occurring from January 2017 to December 2018 in 11 university hospitals that were participating in a national survey of antimicrobial resistance in South Korea [13,14,15]. The 11 hospitals are located in different districts throughout South Korea and have a total of 8762 beds, ranging from 705 to 1779 beds per hospital. The first *Candida*-positive blood samples per patient per species collected [13,14,15], and all *Candida* isolates, were submitted to Chonnam National University Hospital (Gwangju, Korea) for species identification and antifungal susceptibility testing. A total of 829 nonduplicated *Candida* BSI isolates were collected from 807 patients during the 2-year period. Among the 807 patients, 786 had positive blood cultures for a single *Candida* species (338 *C. albicans*, 149 *C. tropicalis*, 147 *C. glabrata* and 111 *C. parapsilosis* and 41 uncommon species), and more than two *Candida* species were isolated simultaneously from 21 patients (mixed candidemia). Information on patient demographics (age, sex, infection origin, admission types), comorbidities, severity of infection (age-adjusted Charlson comorbidity index, severe sepsis, bacteremia), clinical status at the time of positive culture (prior antifungal therapy before blood culture, total parenteral nutrition, surgery within 30 days, neutropenia (less than 500/µL), immunosuppressive therapy, urinary catheter, central venous catheter (CVC)), and therapeutic measures (antifungal therapy and CVC removal) was collected at each sentinel hospital [14,15,16].

### 2.2. Species Identification and Antifungal Susceptibility Testing

*Candida* species were identified based on matrix-assisted laser desorption/ionization–time-of-flight mass spectrometry (Biotyper; Bruker Daltonics, Billerica, MA, USA) with library version 4.0 or sequencing of the D1/D2 domains of the 26S rRNA gene [14,16,17]. In vitro antifungal tests for susceptibility to fluconazole, amphotericin B, and micafungin were performed using the CLSI M27-A3 broth microdilution method. The interpretative guideline in the CLSI document M60 ED1 was used to classify isolates using species-specific clinical break points (CBPs) [18]. If there was no CBP for an antifungal agent for the corresponding species, epidemiological cutoff values (ECVs) were used [19,20].

### 2.3. Definition

Candidemia was defined as the isolation of *Candida* from at least one blood culture [11]; cases of invasive candidiasis without candidemia or colonization were excluded. If the patient with candidemia was hospitalized in the ICU at the time of blood sampling, the candidemia was regarded as ICUAC. Antifungal therapy was defined as the administration of systemic antifungal agents for >72 h; lack of antifungal therapy was defined as no antifungal therapy, or treatment with antifungals for less than 3 days after the date of the first positive blood culture collection [17].

### 2.4. Statistical Analysis

The relationships between mortality and demographic, clinical, and microbiological variables were analyzed by multivariate analysis. Variables with a *p*-value < 0.1 on univariate analysis were included in multivariate analysis. Univariate analyses were based on the chi-squared or Fisher’s exact test, as appropriate, for discrete variables. To identify factors associated with 7-, 30-, and 90-day mortality, backward stepwise Cox regression models were used. Statistical analysis was performed using SPSS (version 26, IBM Corp., Armonk, NY, USA). Statistical significance was determined at a level of *p* < 0.05.

## 3. Results

### 3.1. Species Distribution and Antifungal Resistance

The species distributions and antifungal susceptibility profiles of 829 nonduplicate *Candida* bloodstream isolates from 807 patients are listed in Table 1. The four most common species were *C. albicans* (42.6%), *C. glabrata* (19.2%), *C. tropicalis* (18.8%), and *C. parapsilosis* (13.5%). When we compared the frequencies of *Candida* species between the ICUAC and non-ICUAC groups, *C. tropicalis* was more common in the ICUAC group (23.2% vs. 15.0%, *p* = 0.003). Resistance to fluconazole and micafungin was found in 4.2% (34/813) and 0.2% (2/813) of the *Candida* isolates, respectively. Fluconazole resistance was detected in 6.3% (ICUAC 9.2% vs. non-ICUAC 4.3%) of *C. glabrata* isolates, 1.3% (1.1% vs. 1.5%) of *C. tropicalis* isolates, 3.6% (8.2% vs. 0%, *p* = 0.034) of *C. parapsilosis* isolates, and only 0.3% (0% vs. 0.5%) of *C. albicans* isolates. Overall, fluconazole resistance was more frequent in ICU settings than in non-ICU settings (6.1% vs. 2.5%, *p* = 0.010). Resistance to amphotericin B (exceeding ECV = 2 mg/L) was detected in only one *Candida haemulonii* ICU isolate.

### 3.2. Baseline Characteristics

Among the 807 patients, 370 (45.8%) were classified as having ICUAC. Figure 1 shows the baseline characteristics of all 807 patients and the 745 (92.3%) patients with single-species candidemia attributable to the four common *Candida* species. Several characteristics were more frequently associated with a particular *Candida* species. When ICUAC was compared to non-ICUAC by species, severe sepsis (*C. albicans*, *C. tropicalis* and *C. parapsilosis*), bacteremia (*C. albicans*, and *C. glabrata*), prior surgery (*C. parapsilosis*), urinary catheter placement (all four species), and CVC placement (*C. albicans*, *C. tropicalis* and *C. glabrata*) were frequently associated with ICUAC, in addition to various comorbidities. Of note, ICUAC had a stronger association with lack of antifungal therapy than non-ICUAC (38.9% vs. 31.8%, *p* = 0.035), especially for *C. glabrata* candidemia (ICUAC 55.2% vs. non-ICUAC 32.6%, *p* = 0.007). Additionally, *C. glabrata* ICUAC had a stronger association with lack of antifungal therapy (55.2%) than ICUAC caused by other species (30.6–36.7%, all *p* < 0.05).

### 3.3. Mortality Rate

The cumulative all-cause mortality rates of 807 patients with candidemia at 7, 30, and 90 days were 21.4%, 36.4%, and 43.2%, respectively. The mortality rates in all 370 patients with ICUAC were approximately twofold higher than those in all 437 patients with non-ICUAC at 7 days (2.3 fold, 31.1%/13.3%), 30 days (1.9 fold, 49.5%/25.4%), and 90 days (1.9 fold, 57.8%/30.9%). Figure 2 shows the 7-, 30-, and 90-day mortality rates in patients with ICUAC and non-ICUAC due to the four most common species. Patients with candidemia due to *C. tropicalis* and *C. parapsilosis* showed significantly higher and lower mortality rates, respectively, than all other patients at all observation days (*C. tropicalis*, *p* = 0.002, 0.003, and <0.001; *C. parapsilosis*, *p* = 0.015, 0.001, and <0.001 at 7, 30, and 90 days, respectively, Figure 2A). For the ICUAC group, significantly higher (*C. tropicalis*) and lower (*C. parapsilosis*) species-specific mortality rates were found at only 90 days (*p* = 0.012 and 0.010, respectively); significant species-specific differences in mortality were not observed at 7 and 30 days. Additionally, mortality tended to be higher in patients with *C. glabrata* ICUAC than in those with *C. albicans* ICUAC at 7 and 30 days (Figure 2B). Significantly higher 7-day mortality was observed in patients with *C. tropicalis* non-ICUAC (*p* = 0.014), whereas lower 30- and 90-day mortality rates were found in patients with *C. parapsilosis* non-ICUAC (*p* = 0.006 and 0.001, respectively) (Figure 2C). When ICUAC-associated mortality due to a given *Candida* species was compared with non-ICUAC-associated mortality due to the same species on a given day, mortality was increased approximately twofold (1.7–2.1) in *C. albicans* and *C. tropicalis* candidemic patients at 7, 30, and 90 days and more than threefold in *C. glabrata* candidemic patients at 7 days (3.2 fold) and *C. parapsilosis* candidemic patients at 7, 30 and 90 days (3.1–3.2 fold) (Figure 2D).

### 3.4. Predictors of Mortality

Table 2 summarizes the results of a multivariate analysis of independent predictors of 7-, 30-, and 90-day mortality in all candidemic patients and patients with candidemia due to the four most common species. *C. tropicalis* as a causative agent was an independent predictor of higher mortality in all candidemic patients (odds ratios (ORs), 1.82 at 7 days, 1.45 at 30 days, and 1.43 at 90 days), whereas *C. parapsilosis* was a predictor of lower mortality (OR, 0.62 at 90 days). Several clinical variables were also independent predictors of mortality in all candidemic patients. Among those variables, lack of antifungal therapy was the only predictive factor related to 7-, 30-, and 90-day mortality in patients with candidemia due to all four *Candida* species; the ORs for 7-day mortality due to each type of candidemia were higher than those for 30- or 90-day mortality, especially for candidemia caused by *C. glabrata* (ORs, 45.86 at 7 days, 3.26 at 30 days, and 2.40 at 90 days, respectively). ICU admission was an independent predictor of mortality in candidemia caused by *C. glabrata* (OR, 2.48 at 7 days, 2.15 at 30 days, and 2.07 at 90 days; all *p* < 0.05) and *C. parapsilosis* (OR, 11.54 at 7 days, 6.93 at 30 days, and 6.06 at 90 days; all *p* < 0.05). Of note, fluconazole resistance was an independent predictor of 30- and 90-day mortality due to only *C. glabrata* candidemia (OR, 2.80 at 30 days and 2.84 at 90 days; all *p* < 0.05). Azole monotherapy was protective against mortality in candidemia due to *C. albicans*, *C. tropicalis*, and *C. parapsilosis*.

Table 3 shows the independent predictors of 7-, 30-, and 90-day mortality due to ICUAC and non-ICUAC; these factors varied according to species and observation time. Lack of antifungal therapy was predictive of mortality due to ICUAC and non-ICUAC caused by the four species, except for non-ICUAC due to *C. parapsilosis*. Notably, lack of antifungal therapy was predictive of mortality (7-, 30-, and 90 day) due to ICUAC caused by *C. albicans*, *C. tropicalis*, and *C. glabrata*, and the ORs for 7-day mortality due to ICUAC caused by *C. albicans* (OR 18.33), *C. tropicalis* (OR 10.52) and *C. glabrata* (OR 21.30) were higher than those for 30- and 90-day mortality (ORs, 2.72–6.90). Azole monotherapy was an independent factor protective against 30- and 90-day mortality due to ICUAC caused by *C. albicans*, *C. tropicalis*, and *C. parapsilosis* but not ICUAC caused by *C. glabrata*. Fluconazole resistance was a predictor of 90-day mortality due to *C. glabrata* ICUAC (OR, 5.14, *p* = 0.002). Other than the antifungal therapy variables, most risk or protective factors for mortality due to ICUAC caused by *C. tropicalis*, *C. glabrata*, and *C. parapsilosis* were not predictive of mortality due to non-ICUAC caused by the same species. By contrast, most predictors of mortality due to *C. albicans* ICUAC (severe sepsis, urinary catheter placement, CVC placement and CVC removal) were also predictive of mortality due to *C. albicans* non-ICUAC.

## 4. Discussion

Mortality from candidemia is reportedly associated with several clinical, microbiological, and host-related factors [10,21,22,23,24,25]. Although patient-related variables have consistently been reported as mortality predictors in candidemic patients, not all *Candida* species-related variables have been identified [10,25]. Here, we showed that mortality and the predictors of mortality associated with ICUAC are distinct from those associated with non-ICUAC and are mediated by *Candida* species and observation time (7, 30, and 90 days). Lack of antifungal therapy was the strongest predictor of 7-day mortality due to ICUAC caused by *C. albicans*, *C. tropicalis*, and *C. glabrata* compared with 30- and 90-day mortality. Our data show for the first time that ICU admission and fluconazole resistance are independent predictors of mortality due to *C. glabrata* candidemia, the most common NAC candidemia in many geographic areas [4]. More than half of the patients with *C. glabrata* ICUAC did not receive antifungal therapy for more than 3 days, highlighting the need for early diagnosis of ICUAC.

Few studies have compared the species distribution and resistance profiles of bloodstream *Candida* isolates from ICUAC versus non-ICUAC patients [12,26]. The most common *Candida* species recovered from 11 hospitals over the 2-year period was *C. albicans* (42.6%), followed by *C. glabrata* (19.2%), *C. tropicalis* (18.8%), and *C. parapsilosis* (13.5%), reflecting the recent trends of an increase in the number of *C. glabrata* candidemia cases and a decrease in the number of *C. parapsilosis* candidemia cases in South Korean hospitals [27]. In line with the results of a previous large study using SENTRY Antimicrobial Surveillance Program (2008–2009) data [26], we found that almost half of the candidemia cases developed in ICU patients, and *C. tropicalis* candidemia was more common in ICUAC than non-ICUAC patients. In contrast to the previously reported SENTRY data [26], we found that ICUAC isolates exhibited more resistance to fluconazole than non-ICUAC isolates, possibly because of higher fluconazole resistance in *C. parapsilosis* (ICUAC vs. non-ICUAC, 8.2% vs. 0.0%) and *C. glabrata* (9.2% vs. 4.3%) isolates from ICU patients than in non-ICU-related isolates.

*C. tropicalis* candidemia is associated with a worse outcome than *C. albicans* candidemia [10,21,23,28], whereas *C. parapsilosis* and *C. glabrata* candidemia are associated with better outcomes [10,29,30]. *C. tropicalis* candidemia is often found in ICU patients, especially in those with malignancies [22,29], and *C. tropicalis* might be related to the potential of virulence factors exhibited by this species, such as adhesion to different host surfaces, biofilm formation, infection and dissemination, and enzyme secretion [31]. By contrast, the source of *C. parapsilosis* candidemia was more likely related to removable focuses, such as the CVCs or other intravascular devices, and the lower virulence of *C. parapsilosis* compared to other *Candida* species may be a reason for the lower mortality rates observed [29]. Our data are in agreement with this notion, showing that *C. tropicalis* candidemia was an independent predictive factor for mortality at all time points (7, 30, and 90 days), but *C. parapsilosis* candidemia was associated with a lower 90-day mortality rate. In the ICU setting, significant species-specific differences in 7- and 30-day mortality were not found, but the mortality rate of *C. glabrata* ICUAC tended to be higher than that of *C. albicans* ICUAC at 7 and 30 days. The reason is unclear, but it could be, in part, explained by the fact that the predictors of mortality due to NAC ICUAC were different from those of mortality due to non-ICUAC caused by the same species, unlike those of *C. albicans* candidemia, which may contribute to increased mortality due to ICUAC compared with mortality due to non-ICUAC caused by NAC. Previous reports also showed that the 30-day mortality rates in patients with *C. glabrata* candidemia are 21.3–48.6%, but they can reach 50–60% among ICU patients [10,15]. This study showed that ICU admission was independently associated with mortality due to candidemia caused by *C. glabrata* and *C. parapsilosis* but not *C. albicans* or *C. tropicalis*. The mortality rates of *C. albicans* and *C. tropicalis* ICUAC were approximately twofold higher (1.7–2.1) than those of non-ICUAC on all observation days, whereas the mortality rates of *C. glabrata* (7 days) and *C. parapsilosis* (7, 30, and 90 days) ICUAC were more than threefold higher than those of non-ICUAC. Collectively, our data suggest that mortality due to candidemia and predictors of mortality are affected not only by the NAC species but also by the ICU setting.

Although there is broad consensus that patients who do not receive antifungal therapy have a higher risk of mortality due to candidemia [10,29,32], it is unclear which *Candida* species contribute to this increased mortality due to the lack of antifungal therapy for ICUAC or non-ICUAC. Our findings showed that lack (less than 3 days) of antifungal therapy was the only predictor of mortality in patients with ICUAC and non-ICUAC caused by four common *Candida* species, except for *C. parapsilosis* in non-ICUAC. In particular, it was the strongest predictor of 7-day mortality due to ICUAC caused by *C. albicans* (OR, 18.33), *C. tropicalis* (OR, 10.52), and *C. glabrata* (OR, 21.30) compared with 30- and 90-day mortality (OR, 2.72–6.90). We found that lack of antifungal therapy was more common in ICUAC (38.9%) than in non-ICUAC (31.8%) patients, which may, in part, be a result of earlier death of critically ill ICU patients. Interestingly, lack of antifungal therapy was more frequently found in patients with *C. glabrata* ICUAC (55.2%) than in those with ICUAC caused by other species (30.6–36.7%). The reason for this is unknown; however, previous studies have shown a significantly longer time-to-blood culture positivity for *C. glabrata* (e.g., 61.3 h for *C. glabrata* vs. 25.6 h for the other *Candida* species; *p* < 0.001) [33,34]. Given that non-culture-based diagnostic methods are seldom used in ICU patients in South Korea, this may lead to a delay in initiating antifungal treatment. A longer time-to-positivity duration in candidemic patients may have an impact on mortality [34,35,36]. The recognition of *C. glabrata* candidemia is frequently delayed, resulting in dramatic clinical deterioration and death of critically ill ICU patients [37], which might explain the higher 7-day mortality rate of *C. glabrata* ICUAC. Therefore, these results highlight the importance of the exploration studies of early risk management of ICUAC, such as prophylactic antifungal therapy, biomarker-based preemptive therapy and risk-based empirical therapy in patients with ICUAC, especially *C. glabrata* ICUAC [38,39].

In this study, *C. glabrata* was the only species for which fluconazole resistance was independently related to 30- and 90-day mortality due to candidemia (also 90-day mortality due to ICUAC). This finding is supported by a recent multicenter Korean study that showed that candidemic patients with fluconazole-resistant *C. glabrata* had higher mortality rates [15]. In that study, the cumulative mortality rates of candidemia caused by fluconazole-resistant *C. glabrata* isolates increased over time (day 30 (60.9%) and day 90 (78.1%)); these rates were significantly higher than those in patients with fluconazole-susceptible dose-dependent isolates (36.4% and 43.8%, respectively). Additionally, in that study, appropriate antifungal therapy was the only factor independently associated with favorable outcomes [15]. Our findings also confirmed that azole monotherapy may not promote favorable outcomes of *C. glabrata* candidemia, in contrast to the other three common species. Collectively, these results highlight that continued epidemiological surveillance is important, and future efforts should be directed toward developing new rapid diagnostic techniques—including techniques for candidemia detection, species identification, and antifungal susceptibility testing—to enable the timely administration of appropriate antifungal therapy.

## 5. Conclusions

This is, to the best of our knowledge, the first multicenter study to reveal dynamics in mortalities and mortality-predictors of candidemia according to different *Candida* species and ICU admission. Above all, we highlight that the lack of antifungal therapy is the strong 7-day mortality-predictor of ICUAC, and it is more frequently found in patients with *C. glabrata* ICUAC than in those with ICUAC caused by other species, underscoring the importance of early etiologic diagnosis. Given that the *Candida* species-related mortality rates and mortality predictors of ICUAC are quite distinct from those of non-ICUAC, continued epidemiological surveillance is needed to identify possible changes in the species distribution and antifungal resistance patterns of *Candida* bloodstream isolates from ICU patients.

## Figures and Tables

**Figure 1 jof-07-00597-f001:**
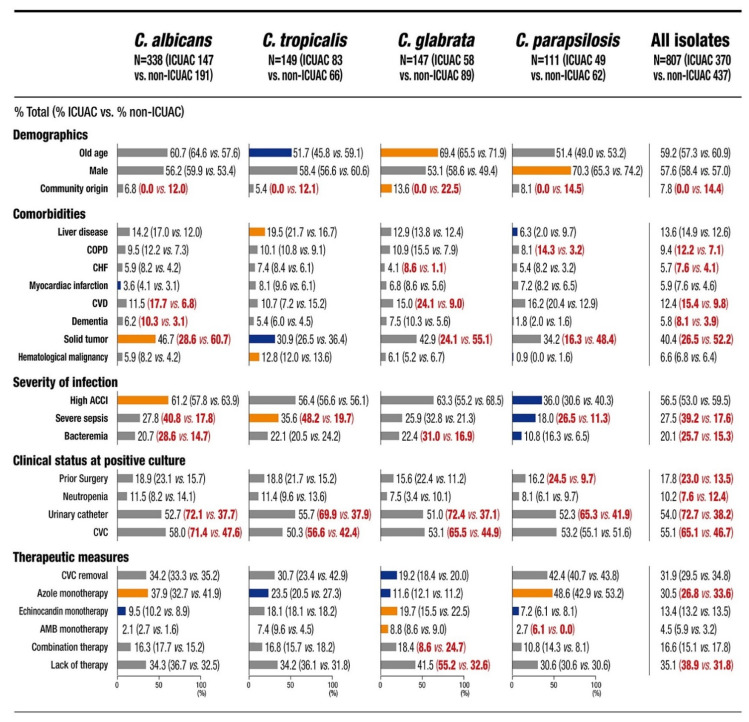
Baseline characteristics of the patients with intensive care unit-associated candidemia (ICUAC) versus non-ICUAC by *Candida* species and all species. The proportions of each variable (%) are indicated in the bar chart, and the statistical significance between a specific *Candida* species and all other *Candida* species within a given category (*p* < 0.05) is represented by colored bars (orange, more frequent; blue, less frequent). The statistical significance (*p* < 0.05) of each variable between ICUAC versus non-ICUAC within a given category is represented by red numbers. Abbreviations: COPD, chronic obstructive pulmonary disease; CHF, congestive heart failure; CVD, cerebrovascular disease; ACCI, age-adjusted Charlson comorbidity index; CVC, central venous catheter; AMB, amphotericin B.

**Figure 2 jof-07-00597-f002:**
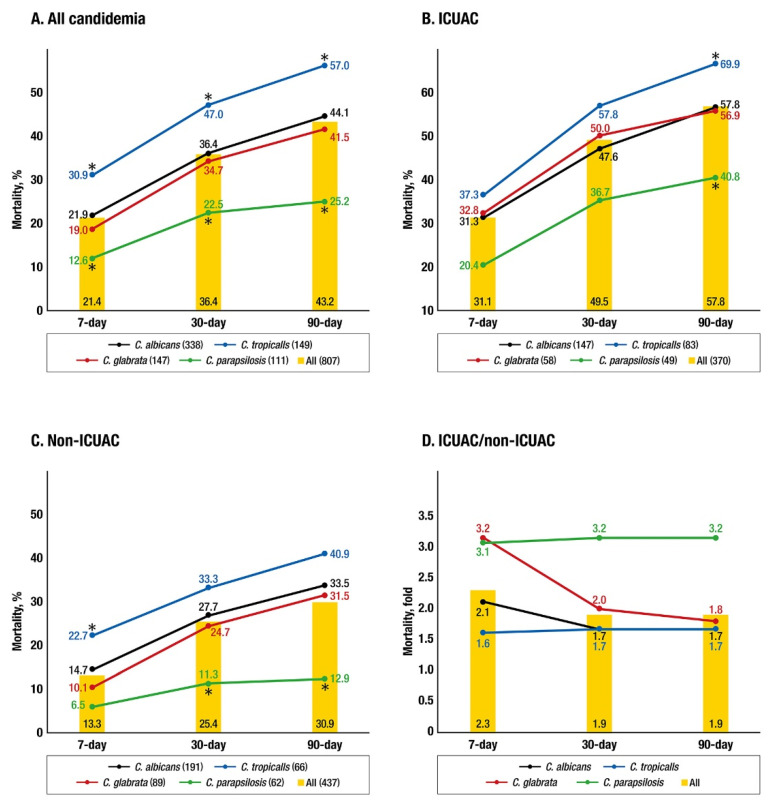
The cumulative 7-, 30-, and 90-day mortality rates in patients with all candidemia (338 *C. albicans*, 149 *C. tropicalis*, 147 *C. glabrata*, 111 *C. parapsilosis*, and 807 all) (**A**), intensive care unit-associated candidemia (ICUAC) (147 *C. albicans*, 83 *C. tropicalis*, 58 *C. glabrata*, 49 *C. parapsilosis*, and 370 all) (**B**), and non-ICUAC (191 *C. albicans*, 66 *C. tropicalis*, 89 *C. glabrata*, 62 *C. parapsilosis*, and 437 all) (**C**). Fold changes in the ICUAC mortality rates compared to the non-ICUAC mortality rates, stratified by four common *Candida* species and all candidemia (**D**). Mortality due to all candidemia within a given category is represented by yellow-colored bars. Asterisks indicate that the mortality rate in patients with candidemia due to a specific *Candida* species was significantly different to those in all other patients within a given category (*p* < 0.05).

**Table 1 jof-07-00597-t001:** Species distribution and antifungal susceptibilities of 829 *Candida* bloodstream isolates from 370 ICU and 437 non-ICU patients in 11 hospitals over a two-year period.

Species	No (%) of Isolates			No. (%) Fluconazole Resistance ^1^			No. (%) Micafungin Resistance ^1^		
	Total	ICU	Non-ICU	Total	ICU	Non-ICU	Total	ICU	Non-ICU
*C. albicans*	353 (42.6)	156 (40.7)	197 (44.2)	1 (0.3)	0 (0.0)	1 (0.5)	0 (0.0)	0 (0.0)	0 (0.0)
*C. glabrata*	159 (19.2)	65 (17.0)	94 (21.1)	10 (6.3)	6 (9.2)	4 (4.3)	1 (0.6)	1 (1.5)	0 (0.0)
*C. tropicalis*	156 (18.8)	89 (23.2) ^2^	67 (15.0)	2 (1.3)	1 (1.1)	1 (1.5)	1 (0.6)	0 (0.0)	1 (1.5)
*C. parapsilosis*	112 (13.5)	49 (12.8)	63 (14.1)	4 (3.6)	4 (8.2) ^2^	0 (0.0)	0 (0.0)	0 (0.0)	0 (0.0)
*C. krusei*	15 (1.8)	10 (2.6)	5 (1.1)	15 (100)	10 (100)	5 (100)	0 (0.0)	0 (0.0)	0 (0.0)
*C. lusitaniae*	9 (1.1)	5 (1.3)	4 (0.9)	1 (11.1)	1 (20.0)	0 (0.0)	0 (0.0)	0 (0.0)	0 (0.0)
*C. guilliermondii*	8 (1.0)	1 (0.3)	7 (1.6)	1 (12.5)	1 (100)	0 (0.0)	0 (0.0)	0 (0.0)	0 (0.0)
*C. dubliniensis*	1 (0.1)	1 (0.3)	0 (0.0)	0 (0.0)	0 (0.0)	-	0 (0.0)	0 (0.0)	-
Others ^3^	16 (1.9)	7 (1.8)	9 (2.0)	NA	NA	NA	NA	NA	NA
Total	829 (100) ^4^	383 (100)	446 (100)	34 (4.2)	23 (6.1) ^2^	11 (2.5)	2 (0.2)	1 (0.3)	1 (0.2)

Abbreviations: ICU, intensive care unit; NA, non-applicable. ^1^ Total proportion of isolates for which the MICs exceed the current Clinical and Laboratory Standards Institute (CLSI) species-specific breakpoints or epidemiological cutoff values [18,19,20]. Total resistance to fluconazole was evident in 4.2% (34/813), 6.1% (23/376), and 2.5% (11/437) of the total, ICU, and non-ICU isolates, respectively; resistance to micafungin was evident in 0.2% (2/813), 0.3% (1/376), and 0.2% (1/437) of the total, ICU, and non-ICU isolates, respectively. ^2^
*p*-value < 0.05, ICU vs. non-ICU. ^3^ Other *Candida* species, for which CLSI breakpoints or epidemiological cutoff values are absent, include 7 isolates (2 *C. haemulonii*, 2 *C. pelliculosa*, 1 *C. fabianii*, 1 *C. lipolytica*, and 1 *C. utilis*) from ICU patients and 9 isolates (2 *C. intermedia*, 2 *C. orthopsilosis*, 1 *C. haemulonii*, 1 *C. pelliculosa*, 1 *C. auris*, 1 *C. metapsilosis*, and 1 *C. norvegensis*) from non-ICU patients. ^4^ A total of 829 non-duplicate *Candida* bloodstream isolates from 807 patients. Only one *Candia* species was recovered from 786 patients, but more than two different *Candida* species were recovered from 21 patients (9 *C. albicans* + *C. glabrata*, 3 *C. albicans* + *C. tropicalis*, 2 *C. tropicalis* + *C. krusei*, 1 *C. albicans* + *C. guilliermondii,* 1 *C. albicans* + *C. lusitaniae* + *C. intermedia*, 1 *C. albicans* + *C. dublinensis*, 1 *C. tropicalis* + *C. glabrata*, 1 *C. tropicalis* + *C. lusitaniae*, 1 *C. glabrata* + *C. parapsilosis*, and 1 *C. glabrata* + *C. guilliermondii*).

**Table 2 jof-07-00597-t002:** Multivariate analysis of predictive factors related to 7-, 30- and 90-day mortalities of all candidemia patients according to the *Candida* species.

Species (No. of Isolates) and Variables ^1^	7-Day		30-Day		90-Day	
	OR (95% CI)	*p*-Value	OR (95% CI)	*p*-Value	OR (95% CI)	*p*-Value
**All (807)**						
Lack of antifungal therapy	14.58 (9.77–21.76)	<0.001	4.71 (3.71–5.98)	<0.001	3.74 (3.00–4.65)	<0.001
CVC placement	2.03 (1.40–2.95)	<0.001	1.59 (1.21–2.08)	0.001	1.54 (1.20–1.98)	0.001
Urinary catheter placement	1.87 (1.29–2.70)	0.001	1.97 (1.47–2.63)	<0.001	2.10 (1.61–2.74)	<0.001
Candidemia due to *C. tropicalis*	1.82 (1.28–2.57)	0.001	1.45 (1.10–1.91)	0.008	1.43 (1.11–1.84)	0.006
Severe sepsis	1.78 (1.30–2.45)	<0.001	1.92 (1.49–2.46)	<0.001	2.04 (1.62–2.57)	<0.001
Prior surgery	0.55 (0.36–0.86)	0.009	-	-	-	-
Azole monotherapy	0.17 (0.10–0.30)	<0.001	0.45 (0.33–0.60)	<0.001	0.50 (0.39–0.66)	<0.001
Hematologic malignancies	-	-	1.52 (1.01–2.29)	0.04	-	-
ICU admission	-	-	1.43 (1.09–1.86)	0.009	1.39 (1.09–1.76)	0.008
CVC removal	-	-	0.64 (0.45–0.92)	0.01	0.58 (0.41–0.80)	0.001
Candidemia due to *C. parapsilosis*	-	-	-	-	0.62 (0.42–0.93)	0.01
***C. albicans* (338)**						
Lack of antifungal therapy	17.82 (9.32–34.08)	<0.001	5.87 (4.02–8.56)	<0.001	4.76 (3.37–6.70)	<0.001
CVC placement	2.44 (1.38–4.34)	0.002	2.14 (1.39–3.29)	0.001	2.22 (1.50–3.28)	<0.001
Urine catheter use	1.94 (1.13–3.33)	0.01	2.26 (1.48–3.44)	<0.001	2.45 (1.66–3.60)	<0.001
Severe sepsis	-	-	1.80 (1.22–2.67)	0.003	1.95 (1.36–2.78)	<0.001
Prior surgery	0.39 (0.18–0.85)	0.01	-	-	-	-
Azole monotherapy	0.13 (0.06–0.29)	<0.001	0.35 (0.23–0.55)	<0.001	0.42 (0.29–0.60)	<0.001
CVC removal	-	-	0.55 (0.33–0.92)	0.02	0.49 (0.30–0.81)	0.006
***C. tropicalis* (149)**						
Lack of antifungal therapy	8.64 (4.52–16.53)	<0.001	4.19 (2.54–6.89)	<0.001	3.65 (2.32–5.75)	<0.001
CVC placement	2.29 (1.10–4.75)	0.02	-	-	-	-
Urinary catheter placement	2.21 (1.02–4.81)	0.04	2.59 (1.45–4.61)	0.001	2.95 (1.73–5.03)	<0.001
Severe sepsis	-	-	2.18 (1.29–3.67)	0.003	2.39 (1.48–3.85)	<0.001
Azole monotherapy	0.35 (0.14–0.90)	0.02	-	-	0.48 (0.27–0.86)	0.01
Congestive heart failure	-	-	2.48 (1.20–5.12)	0.01	2.10 (1.03–4.28)	0.04
Neutropenia	-	-	2.17 (1.10–4.26)	0.02	2.22 (1.17–4.23)	0.01
***C. glabrata* (147)**						
Lack of antifungal therapy	45.86 (6.19–339.57)	<0.001	3.26 (1.84–5.80)	<0.001	2.40 (1.43–4.01)	0.001
Severe sepsis	-	-	1.93 (1.08–3.43)	0.02	2.30 (1.37–3.86)	0.002
ICU admission	2.48 (1.11–5.51)	0.02	2.15 (1.21–3.83)	0.009	2.07 (1.22–3.49)	0.007
Fluconazole resistance	-	-	2.80 (1.17–6.66)	0.02	2.84 (1.28–6.33)	0.01
***C. parapsilosis* (111)**						
Lack of antifungal therapy	36.30 (4.28–307.86)	0.001	9.18 (2.75–30.61)	<0.001	4.72 (1.51–14.76)	0.008
Urinary catheter placement	-	-	-	-	4.35 (1.13–16.71)	0.03
Azole monotherapy	0.05 (0.01–0.43)	0.006	0.28 (0.10–0.84)	0.02	-	-
ICU admission	11.54 (1.38–96.67)	0.02	6.93 (1.84–26.09)	0.004	6.06 (1.92–19.17)	0.002
CVC removal	-	-	-	-	0.27 (0.08–0.88)	0.02

Abbreviations: OR, odds ratio; 95% CI, 95% confidence interval; ICU, intensive care unit; CVC, central venous catheter. ^1^ Only the variables that were statistically significant (*p* < 0.05) are listed. The results of univariate analysis are listed in Appendix A.

**Table 3 jof-07-00597-t003:** Multivariate analysis of predictive factors related to 7-, 30- and 90-day mortalities of patients with ICUAC and non-ICUAC by the four common *Candida* species.

Species (No. of Isolates) and Variables ^1^	Setting	7-Day		30-Day		90-Day	
		OR (95% CI)	*p*-Value	OR (95% CI)	*p*-Value	OR (95% CI)	*p*-Value
***C. albicans* (ICU 147; non-ICU 191)**							
Lack of antifungal therapy	ICUAC	18.33 (7.65–43.92)	<0.001	6.90 (4.12–11.57)	<0.001	5.47 (3.42–8.75)	<0.001
	Non-ICUAC ^2^	34.64 (7.86–152.68)	<0.001	10.49 (4.68–23.51)	<0.001	3.97 (2.32–6.79)	<0.001
CVC placement	ICUAC	-	-	-	-	1.86 (1.01–3.43)	0.04
	Non-ICUAC	-	-	-	-	2.05 (1.20–3.51)	0.009
Urinary catheter placement	ICUAC	-	-	2.56 (1.25–5.24)	0.01	3.28 (1.65–6.51)	0.001
	Non-ICUAC	-	-	2.71 (1.25–5.84)	0.03	1.79 (1.07–2.99)	0.02
Severe sepsis	ICUAC	-	-	1.69 (1.04–2.75)	0.03	1.62 (1.03–2.56)	0.03
	Non-ICUAC	-	-	-	-	2.57 (1.49–4.43)	0.001
Prior surgery	ICUAC	0.40 (0.17–0.96)	0.04	-	-	-	-
Azole monotherapy	ICUAC	0.11 (0.03–0.35)	<0.001	0.27 (0.14–0.52)	<0.001	0.33 (0.19–0.59)	<0.001
	Non-ICUAC	0.06 (0.01–0.44)	0.006	0.26 (0.11–0.62)	0.002	0.46 (0.27–0.77)	0.004
CVC removal	ICUAC	-	-	-	-	0.49 (0.27–0.88)	0.01
	Non-ICUAC	-	-	-	-	0.34 (0.15–0.80)	0.01
***C. tropicalis* (ICU 83; non-ICU 66)**							
Lack of antifungal therapy	ICUAC	10.52 (3.91–28.25)	<0.001	4.01 (2.15–7.47)	<0.001	3.79 (2.14–6.74)	<0.001
	Non-ICUAC	7.74 (2.59–23.17)	<0.001	4.54 (1.87–11.00)	0.001	6.02 (2.45–14.81)	<0.001
Severe sepsis	ICUAC	-	-	2.71 (1.42–5.19)	0.004	2.16 (1.14–4.11)	0.002
Congestive heart failure	ICUAC	3.93 (1.17–13.15)	0.02	3.53 (1.43–8.69)	0.006	3.16 (1.31–7.63)	0.01
Neutropenia	ICUAC	-	-	2.50 (1.06–5.88)	0.03	2.50 (1.09–5.73)	0.01
Azole monotherapy	ICUAC	-	-	0.21 (0.08–0.59)	0.003	0.20 (0.08–0.46)	0.03
***C. glabrata* (ICU 58; non-ICU 89)**							
Lack of antifungal therapy	ICUAC	21.30 (2.79–162.83)	0.003	2.72 (1.12–6.62)	0.02	3.41 (1.59–7.32)	0.002
	Non-ICUAC	-	-	5.02 (1.82–13.88)	0.002	-	-
Urinary catheter placement	ICUAC	-	-	6.81 (1.71–27.06)	0.006	-	-
Solid tumor	ICUAC	3.64 (1.38–9.57)	0.009	2.86 (1.23–6.70)	0.01	-	-
Chronic kidney disease	ICUAC	-	-	4.05 (1.45–11.31)	0.007	-	-
Fluconazole resistance	ICUAC	-	-	-	-	5.14 (1.80–14.65)	0.002
***C. parapsilosis* (ICU 49; non-ICU 62)**							
Lack of antifungal therapy	ICUAC	-	-	22.77 (4.17–124.26)	<0.001	12.90 (4.16–39.98)	<0.001
Urinary catheter placement	ICUAC	-	-	-	-	4.81 (1.25–18.48)	0.02
Azole monotherapy	ICUAC	-	-	0.13 (0.03–0.50)	0.003	0.19 (0.06–0.55)	0.002
Male	ICUAC	0.26 (0.07–0.95)	0.04	-	-	-	-
Old age (≥65 yrs)	ICUAC	9.96 (1.25–79.60)	0.03	-	-	-	-

Abbreviations: OR, odds ratio; 95% CI, 95% confidence interval; ICUAC, intensive care unit-associated candidemia; CVC, central venous catheter. ^1^ Only the variables that were statistically significant (*p* < 0.05) are listed. The results of univariate analysis are listed in Appendix A. ^2^ Only mortality-predictive factors of non-ICUAC that were commonly found in both ICUAC and non-ICUAC are listed here. All independent predictive factors related to 7-, 30-, and 90-day mortalities of non-ICUAC are listed in Appendix A.

## Data Availability

All data generated or analyzed in this study are included in this published article, and the datasets are available from the corresponding author within the limits imposed by ethical and legal dispositions.

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
