# Peer review of "Dynamics and Predictors of Mortality Due to Candidemia Caused by Different Candida Species: Comparison of Intensive Care Unit-Associated Candidemia (ICUAC) and Non-ICUAC"

_jof, 2021, doi:10.3390/jof7080597_

Round 1
Reviewer 1 Report
The manuscript describes a study in which predictors of mortality caused by Candida are evaluated in different hospitals from South Korea. The study is well conducted and the results are well reported and explained, supported by figures and tables. However, there are some minor comments that should be addressed before publishing.
Minor comments:
- Sections 2.1 and 2.2 are exactly the same thing. Please remove one of them
- Line 110: in vitro should be in italics
- In section 2.5 authors say they performed univariate analysis, but in section 3.4 they talk about multivariate analysis. Please specify.
- In section 3.4, C. parapsilosis was reported as a protective factor for 90-day mortality. How is an infection supposed to be protective?
- In the discussion, I missed some referencing to literature. The authors describe which species are related with higher mortality based on the predictions they obtained from multivariate analysis, but they did not justify with a scientific evidence. For example, “these species were found to be related with a worse outcome, this could be explained by the fact that these species are multidrug resistant according to literature” or something like that.
- I missed a concluding paragraph to sum up everything that had been reported.
- I also missed some “future work” related content. It’s been a good work, but the purpose of it remains unclear. The authors report the morbility caused by fungal species for what?
Author Response
- Sections 2.1 and 2.2 are exactly the same thing. Please remove one of them
Answer: We have deleted the repeated section and the numbering of section had been revised accordingly, as the reviewer suggested (Lines 76-95).
- Line 110: in vitro should be in italics
Answer: We have converted it in italic, as the reviewer suggested (Line 100).
- In section 2.5 authors say they performed univariate analysis, but in section 3.4 they talk about multivariate analysis. Please specify.
Answer: The sentences have been included according to the reviewer’s comment, and now reads “The relationships between mortality and demographic, clinical, and microbiological variables were analyzed by multivariate analysis. Variables with a p-value < 0.1 on univariate analysis were included in multivariate analysis.” (Lines 115-117).
- In section 3.4, C. parapsilosis was reported as a protective factor for 90-day mortality. How is an infection supposed to be protective?
Answer: The sentences have been modified to avoid causing confusion, and now reads “C. tropicalis as a causative agent was an independent predictor of higher mortality in all candidemic patients (odds ratios [ORs], 1.82 at 7 days, 1.45 at 30 days, and 1.43 at 90 days), whereas C. parapsilosis was a predictor of lower mortality (OR, 0.62 at 90 days).”, according to the reviewer’s comment (Lines 198-201).
- In the discussion, I missed some referencing to literature. The authors describe which species are related with higher mortality based on the predictions they obtained from multivariate analysis, but they did not justify with a scientific evidence. For example, “these species were found to be related with a worse outcome, this could be explained by the fact that these species are multidrug resistant according to literature” or something like that.
Answer: The issues and references on higher or lower mortality of candidemia due to C. tropicalis, C. parapsilosis and C. glabrata have been included in line with the reviewer’s comment (Lines 284-291; 300-302, references 22, 29, 31, 29, 10, 15).
- I missed a concluding paragraph to sum up everything that had been reported.
Answer: We have added a concluding paragraph according to the reviewer’s comment (Lines 350-360).
- I also missed some “future work” related content. It’s been a good work, but the purpose of it remains unclear. The authors report the morbility caused by fungal species for what?
Answer: We have added the content of “future work”, in line with the reviewer’s comment. (Lines 330-333, 346-349, 356-360).
Reviewer 2 Report
This is an interesting paper defining all candidemia cases in 11 hospitals from 2017 to 2018 in South Korea.
The article is well-written, cohesive and original considering high diffusion of multi drug resistant microorganisms.
However I have some comments for authors.
Method section: selection criteria is also important to ensure diversity of subset is reflective of the diversity of the population overall. How were the representative strains selected?
Add in discussion section a sentence to provide an explanation of the importance and relevance of the study to the field.
Author Response
- Method section: selection criteria is also important to ensure diversity of subset is reflective of the diversity of the population overall. How were the representative strains selected?
Answer: Selection criteria have been added with references (our previous studies), according to the reviewer’s comment (Lines 77-84).
- Add in discussion section a sentence to provide an explanation of the importance and relevance of the study to the field.
Answer: Sentences to provide an explanation of the importance and relevance of our study to the field has been added, according to the reviewer’s comment (Lines 330-333, 351-360).